# Energy Availability in Male and Female Elite Wheelchair Athletes over Seven Consecutive Training Days

**DOI:** 10.3390/nu12113262

**Published:** 2020-10-25

**Authors:** Thomas Egger, Joelle Leonie Flueck

**Affiliations:** 1Laboratory of Exercise and Health, Department of Health Sciences and Technology, Swiss Federal Institute of Technology (ETH) Zurich, 8092 Zurich, Switzerland; thomas.patrick.egger@gmail.com; 2Institute for Sports Medicine, Swiss Paraplegic Centre, 6207 Nottwil, Switzerland

**Keywords:** sports nutrition, resting energy expenditure, spinal cord injury, RED-S, paralympic, energy availability, female athlete triad, wheelchair athlete, macronutrient intake

## Abstract

Background: Low energy availability (LEA) is a major problem as athletes often restrict their energy intake. It has been shown that LEA occurs often in female and endurance athletes and in athletes from weight-sensitive or aesthetic sports. The purpose of this study was to investigate energy availability (EA) in elite wheelchair athletes. Methods: Fourteen elite wheelchair athletes (8 males; 6 females) participated. Data were collected using a weighed seven-day food and training diary to estimate energy intake and exercise energy expenditure. Resting energy expenditure and body composition were measured, whereas energy balance (EB) was calculated. Results: Measured over 7 days, EA was significantly different (36.1 ± 6.7 kcal kg^−1^ FFM day^−1^) in male compared to female (25.1 ± 7.1 kcal kg^−1^ FFM day^−1^) athletes (*p* < 0.001). From all analyzed days, LEA occurred in 73% of the days in female athletes and in 30% of the days in male athletes. EB was positive in male athletes (+169.1 ± 304.5 kcal) and negative (−288.9 ± 304.8 kcal) in female athletes. Conclusions: A higher prevalence of LEA was found in female compared to male athletes. A higher energy intake would be recommended to meet energy needs and to maximize training adaptation.

## 1. Introduction

Energy availability (EA) and low energy availability (LEA) have been researched extensively in the last ten years [1,2]. The first studies focused mainly on female athletes since LEA has been proposed as one of the three main factors leading to the female athlete triad resulting in a decreased bone mineral density and in menstrual dysfunctions [3,4]. More recently, it has been shown that the female athlete triad might be much more complex with more contributing factors [5,6]. Moreover, these health issues have been reported in male athletes as well [2]. Thus, the term of relative energy deficiency in sport (RED-S) was an expansion to describe in more detail the complexity of these health and performance compromising issues caused by an inadequate energy intake in relation to training volume in male and female athletes [7,8].

EA is defined as energy intake (EI) [kcal day^−1^] minus EEE (Exercise Energy Expenditure) [kcal day^−1^] and is relative to the fat-free mass (FFM) [9]. Values at or slightly above 45 kcal kg^−1^ FFM day^−1^ constitute optimal EA, whereas values between 30 kcal kg^−1^ FFM day^−1^ and 45 kcal kg^−1^ FFM day^−1^ are referred as a suboptimal EA. An EA at or below 30 kcal kg^−1^ FFM day^−1^ is defined as LEA [2]. Thus, LEA arises if nutritional intake does not cover the energy demands of FFM regarding exercise energy expenditure. Some physiological dysfunctions might evolve, such as impaired cognition, impaired cardiovascular health, poor bone health, reproductive disturbances, low immunity and some gastrointestinal disturbances [2,10]. To achieve long-term health, optimal EA (>45 kcal kg^−1^ FFM day^−1^) should be reached daily [2].

These cut-off values have been defined for able-bodied athletes but it seems unsure if these would be also applicable to wheelchair athletes with a spinal cord injury (SCI) [11]. Body-related cognition, energy expenditure (EE), gastrointestinal health, bone mineral density, as well as reproductive function might already be affected by the SCI or the disability itself in wheelchair athletes [11,12]. Therefore, it seems complex to study the health-related issues of LEA in this population. Furthermore, LEA might also emerge in a stable energy balance with constant body mass and balanced EI and EE due to adaptive reduction in resting energy expenditure (REE) following chronic LEA [2,13]. As there is currently no specific single assessment available for the detection of LEA that would facilitate diagnosis [12], it seems extremely difficult to detect LEA without the extensive examination of EI, FFM and EEE as well as bone mineral density, REE and blood hormone parameters. In addition, the examination of dietary intake and EEE is susceptible to errors by over- or underestimating EI and EEE [1,14].

In the athletic population, a reduced EA is acceptable for a short-term duration and might be required to reach optimal body composition before a major competition (e.g., the last few weeks) [9]. Therefore, periodization of nutritional approaches (e.g., increase in fat oxidation, training with high or low carbohydrate (CHO) availability, weight loss/gain at different training phases) is as important as training periodization [15,16,17]. Therefore, it would be useful to study a microcycle of seven consecutive days to investigate the nutritional intake on days with different training loads. 

The prevalence and health consequences of LEA in wheelchair athletes have not been investigated in detail yet, although these athletes are considered to be a population at a high-risk of LEA [5,11,12]. Therefore, research is warranted to examine EA in athletes with SCI in a first step using the same cut-off values as for able-bodied athletes. In a second and third step, the analysis of the health consequences and the performance impairment (i.e., risk of injury, fracture risk, immune function, loss of training days, etc.) might be studied in order to develop a more specific tool or cut-off values for this special population (if needed). Consequently, the purpose of this first study was to observe EA in male and female wheelchair athletes to detect whether this cohort develops LEA or not. In a secondary evaluation of the data, we aimed to investigate the nutritional habits of the athletes pre- and post-exercise to observe whether sports nutritional recommendations have been deployed by the athletes and whether the athletes fueled themselves differently on days with a higher or lower training volume. 

## 2. Materials and Methods 

### 2.1. Participants

Swiss elite wheelchair athletes between the ages of 18 and 60 years old were recruited. They had to be a member of the Swiss national team in their wheelchair sports and healthy. Athletes were excluded if medical issues affected their normal training schedule and dietary intake, or if it was not possible to conduct a body composition or a REE measurement. Furthermore, athletes should not have been in a phase of intended weight loss.

Participants had to visit the laboratory of the Institute for Sports Medicine once. At the visit, they provided written informed consent, filled in a health questionnaire and their height and body mass were measured. Subsequently, REE was measured and they underwent dual energy X-ray absorptiometry (DXA) to evaluate body composition. Finally, they were instructed individually in detail by a trained sports nutritionist on how to log the food and training diary. After this visit, the athletes had to record food and beverage intake and training for seven subsequent days. After the data collection, athletes received support from a trained sports nutritionist to further optimize their nutritional intake around training sessions. The study protocol was approved by the local ethical committee (ID No. 2019-00024, Ethikkommission Nordwest- und Zentralschweiz (EKNZ), Basel, Switzerland). 

### 2.2. Body Composition and Resting Energy Expenditure

Measurements for body composition and REE were performed in the morning after an overnight fast and without performing any high-intensity training session in the last 24 h before the visit. Body mass was measured using a wheelchair scale (BIT 650 from Bizerba Busch, Balingen, Germany) and participants wore minimal clothing. The mass of the wheelchair was measured independently and subtracted from the total mass (i.e., athlete + wheelchair). Height was measured with participants in the supine position on a mat with the feet at the wall and the height marked on the mat. The distance from the wall to the mark was measured. 

REE was measured using a metabolic cart (Oxycon Pro, Erich Jaeger GmbH, Hoechberg, Germany). The measurement took place at a room temperature of 20 °C at 59% humidity. The metabolic cart was calibrated before each measurement by volume and gas calibration according to the manufacturer’s guidelines (measuring precision for VO_2_ and VCO_2_ ± 3% or 0.5 L/min). The REE was measured over 20 min in a supine position after a 10-min rest period. The measurement was performed using a breath-by-breath technique with the athletes wearing a mask (Hans-Rudolph mask, Hans Rudolph Inc., Shawnee, KS USA). Data were averaged over 15-s periods. A 3 to 5 min period with steady-state conditions was used for REE determination. REE was calculated by the software of the metabolic cart (JLAB, Jaeger GmbH, Hoechberg, Germany) by using the equation of Weir [18]. In one participant, REE had to be estimated using an equation for SCI [19]. Subsequently, a full-body DXA was conducted with the Lunar iDXA scanner (GE Healthcare Lunar, Madison, WI, USA) by a trained technical assistant of the Institute of Radiology. The iDXA was calibrated daily with phantoms as per manufacturer guidelines. All the scans were undertaken and analyzed using the array mode (encore software, version 17, GE Healthcare, Madison, WI, USA) by the same technician. A trained and experienced radiographer positioned all participants as best as possible to obtain a valid measurement. If they had spasms during the scan, they were repositioned and the scan was repeated. To minimize the risk for hypohydration, athletes were instructed to drink a glass of water beforehand but to refrain from energy intake 12 h prior to testing [20]. In addition, as most wheelchair athletes need to take some medication in the morning, a glass of water seemed appropriate from a practical point of view. The two last meals were recorded through a recall questionnaire. Athletes were advised not to restrict CHO the day before the test procedure in order to obtain normal glycogen stores. The participants wore minimal clothing and their bladders were voided. Measurement precision values in elite wheelchair athletes for fat mass, fat-free mass and bone mineral content were all below 2% [21].

### 2.3. Self-Reported Food and Training Diary

For the evaluation of EI and EEE data, participants provided a self-reported food and training diary, which was recorded over seven consecutive days pre-season during springtime. All participants were introduced to the recording process of the diaries by a trained nutritionist. They were instructed to weigh and record all food items ingested, as well as all beverages including supplements, sport-specific nutrition and sports drinks. In addition, they were advised to provide photographs of the meals at the end of the recording week (application “See what you eat”, Health Revolution Ltd., Kotka, Finland), especially for those meals where weighing was not possible. Approximations for the food weight based on the photographs were made using the menuCH photobook [22]. The food diary was analyzed for every participant by the same nutritionist with nutrition database software (PRODI 6.7 expert Swiss, Nutri-Science GmbH, Stuttgart, Germany). To calculate whether the athletes followed the sports nutritional guidelines pre- and post-training, training nutrition was analyzed separately. 

The training diary contained the duration and intensity of the particular training session. In addition, a brief description of the training session, as well as the heart rate and rate of perceived exertion (RPE) ranging between 6 and 20 on the Borg scale [23], were provided. Training sessions had to be rated with one out of four intensities according to Hottenrott (e.g., recovery, basic endurance, moderate endurance, submaximal to maximal work (interval training)), which the participants already were familiar with [24]. The diary also included the leisure time physical activity questionnaire for spinal cord injured (LTPAQ-SCI) to report non-practice activities [25]. Additionally, participants were asked for their sleep duration in order to correct EE for sleep.

### 2.4. Calculations

Resting energy expenditure measurements were used to calculate EE throughout the day and night (sleep adjusted with physical activity level factor (PAL) = 0.95 [26]). The EEE estimates of the different training sessions were calculated according to published data on EEE in wheelchair athletes (Table 1) for specific sports and intensities [27]. Data from performance testing from each athlete and training zones according to heart rate or power (Watt) were available. To calculate EEE, training duration as well as body mass of the athlete were multiplied with the value for the specific exercise and intensity displayed in Table 1. To be able to estimate total energy expenditure (TEE) in addition to EA, the thermic effect of food (TEF) was calculated according to Jéquier [28]. To calculate the EE of the leisure time physical activity (LTPA), daily duration was multiplied with the EE kg^−1^ body mass (BM) h^−1^ [25]. The EE for mild activities was 1.6 kcal kg^−1^ BM h^−1^ and 2.6 kcal kg^−1^ BM h^−1^ for moderate activities and 3.6 kcal kg^−1^ BM h^−1^ for heavy activities [27]. TEE was calculated using the equation: TEE (kcal d^−1^) = (REE h^−1^ * time awake (h)) + (0.95 * REE h^−1^ * time asleep (h)) + TEF + (EE spent during LTPA) + EEE. The EA was analyzed using (EI − EEE)/kg FFM [12]. The FFM from the DXA scan was used in this equation as well as the energy intake calculated from the food diary. 

### 2.5. Guidelines of Macronutrient Intake

Minimal daily CHO intakes of >3 g kg^−1^ BM, protein intakes of >1.2 g kg^−1^ BM and additionally 20 to 25 g protein 0 to 60 min post-training and a minimum of 1 g kg^−1^ BM carbohydrates the last 4 h pre- and post-training were assumed adequate [29]. No special guidelines were available for para-athletic populations.

### 2.6. Statistical Analysis

Statistical analyses were performed using SPSS version 26 (IBM Corp, Armonk, NY, USA). Data were tested for normality using the Shapiro–Wilk test. All data were normally distributed. Data were presented as mean ± standard deviation. Significance level was set at 0.05. A t-test was used to detect differences between male and female athletes. In order to determine the effect size, Hedge’s g (g) was calculated. The effect was small when g > 0.20, moderate when >0.5 and large when >0.8 [30]. Pearson’s correlation was chosen to detect any relationship between variables.

## 3. Results

In total, 15 wheelchair athletes gave written informed consent to participate in this study—14 completed data collection (Table 2). Most athletes participating in this study were daily wheelchair users (*n* = 13, 7 male, 6 female) due to SCI (*n* = 9, male = 7, female = 2), spina bifida (*n* = 4, male = 1, female = 3) or multiple sclerosis (*n* = 1 female). 

Male athletes displayed LEA on 30% of the days whereas female athletes displayed LEA on 73% of all days (Table 3). Table 4 shows the detailed information on mean EA, EI and TEE. The mean EI per day of 29% of the participants (*n* = 4, male: *n* = 3, female: *n* = 1, 8 out of 14) was at or above 45 kcal kg−1 FFM day−1 and 14% (2 out of 14) were below 30 kcal kg−1 FFM day−1. The effect size for the difference between male and female was large for EI (g = −2.37), TEE (g = −1.38) and energy balance (g = −1.41).

The macronutrient intake over the whole day, as well as the dietary intake before and after training sessions, is displayed in Table 5. The effect size between male and female athletes was large for total CHO intake (g = −7.20), relative CHO intake (g = −4.99), absolute protein intake (g = −3.61), relative protein intake (g = −0.90), absolute fat intake (g = −3.98) and relative fat intake (g = −2.50). Energy balance for the seven days of each participant is displayed in Figure 1.

Between weekdays and the weekend, the relative intake of protein, fat and CHO did not significantly differ (Figure 2). Mean EA for weekdays was 31.4 ± 13.2 kcal kg^−1^ FFM day^−1^ whereas for weekend was 30.4 ± 12.8 kcal kg^−1^ FFM day^−1^ (*p* = 0.74). No significant difference (*p* = 0.53) was found in EI between weekdays (1907 ± 709 kcal) and weekends (1807 ± 722 kcal). Exercise energy expenditure was similar (*p* = 0.27) between weekdays (442 ± 322 kcal) and weekends (360 ± 352 kcal). The Pearson correlation showed a positive correlation between EI and EEE (*p* <0.001). Furthermore, significant correlations between EEE and relative protein (*p* = 0.016), fat (*p* = 0.002) and CHO (*p* = 0.001) as well as EA (*p* = 0.028) were found.

## 4. Discussion

The purpose of this study was to examine EA among elite wheelchair athletes. In total, 93% of the athletes had a weekly mean EA below 45 kcal kg^−1^ FFM day^−1^. In addition, overall mean EA was 31.4 ± 8.7 kcal kg^−1^ FFM day^−1^ for all participants (Table 4). Female athletes showed a mean EA of 25.1 ± 7.1 kcal kg^−1^ FFM day^−1^, whereas male athletes showed a significantly higher mean EA of 36.1 ± 6.7 kcal kg^−1^ FFM day^−1^. Furthermore, of all the days analyzed, female athletes displayed LEA on 73% of all days, whereas LEA in male athletes was reported on 30% of the days. This is in line with the negative energy balance, which resulted in five out of six female and in one out of eight male athletes over the seven days period. These data show the importance of the further analysis of the relationship of LEA and health outcome parameters in wheelchair athletes as well as the investigation of the cut-off values based on data in this population. 

### 4.1. Energy Availability and Energy Intake

The occurrence of LEA in female wheelchair athletes was very high (>70%) compared to male athletes (Table 3). In comparison to able-bodied literature, LEA was shown in 18% of female adolescent Kenyan elite runners [31] and in 20% of female elite runners and triathletes [32]. Furthermore, LEA affected 20% of female varsity volleyball players [33], 31% of female elite long-distance runners [34], 51% of female endurance runners [35] and 52% of female collegiate runners [36]. Thus, the occurrence of LEA seems to be higher in female wheelchair athletes compared to able-bodied female athletes. Therefore, either female wheelchair athletes are more prone to LEA or the cut-off values to classify a wheelchair athlete with LEA might be different compared to able-bodied athletes [12]. In addition, underreporting or energy restriction in order to reduce body weight pre-season might be possible. Thus, more studies are needed to investigate EA in this population as well as to study the relationship between LEA and secondary health parameters to be able to classify LEA in wheelchair athletes more accurately. 

In contrast to female athletes, only one out of eight male wheelchair athletes showed a LEA over the weekly mean (Table 3). This result is lower than in able-bodied counterparts, where the occurrence of LEA ranged between 25% in male elite runners [34], 28% in male competing cyclists [37] and up to 58% in male endurance athletes [35]. This is rather a surprising difference compared to our results, as only two male athletes in our study were from non-endurance sports. It is possible that these male athletes were at an ideal weight and were not aiming to restrict their diet and lose body mass. Furthermore, it is possible, that in pre-season, their training intensity was higher and they were paying attention to adequate energy intake to maximize performance and recovery. Still, one might expect a similar occurrence of LEA in male wheelchair endurance athletes as found in able-bodied endurance athletes as the sport per se and the importance of body weight in endurance sports is not different. This again questions whether cut-off values for LEA must be altered in wheelchair athletes compared to able-bodied athletes. 

In addition to EA, weekly mean energy balance (Figure 1) was also negative in five of six female compared to only one of eight male athletes. Figure 1 also indicates that most of these five females were in a negative energy balance over all seven days. Male athletes mainly stayed within a positive energy balance throughout the week. Their overall reported EI was 2276.3 ± 368.3 kcal d^−1^, which was similar to other wheelchair athletes [38,39] or even 10 to 25% higher [40,41,42]. The reported energy intake of our female wheelchair athletes was extremely low (1377 ± 337 kcal d^−1^). The reported EI in this study was 10 [40,41] to 30% [38,42,43] lower compared to other studies investigating EI in female wheelchair athletes. This could be due to the fact that this study took place pre-season and athletes were possibly restricting their EI to induce weight loss. Additionally, we cannot exclude under-reporting as stated later in the limitations. To summarize, it seemed crucial to educate wheelchair athletes, especially female athletes, in order to make sure that nutritional intake is sufficient to induce optimal training adaptations.

### 4.2. Macronutrient Intake before and after Training Sessions

We evaluated the macronutrient intake over a complete training day as well as before and after training sessions. Male athletes had a daily carbohydrate intake of 4.0 ± 1.0 g kg^−1^ BM and a protein intake of 1.4 ± 0.1 g kg^−1^ BM (Table 5). These macronutrient intakes are within recommendations [29] and comparable to previous results of male wheelchair athletes [40,41,43]. Female athletes instead showed lower CHO (2.4 ± 0.7 g kg^−1^ BM) and protein (1.1 ± 0.3 g kg^−1^ BM) intakes compared to male athletes in this study. In comparison to other cohorts, the carbohydrate and protein intakes were lower (1.6 ± 0.6 g kg^−1^ BM [41,43]) or equal (1.1 g kg^−1^ BM [40]). Again, it seems that those athletes who restricted energy intake without reporting it or due to pre-seasonal intended weight loss. Another reason could be a higher exercise intensity or volume (i.e., pre-season) without adjusting energy intake. Therefore, it seems not surprising that macronutrient intake was lower and LEA occurred more often compared to studies performed on able-bodied individuals. We assume that nutritional counseling in female wheelchair athletes should be of major interest to ensure optimal health and performance. Furthermore, increasing CHO intake over the whole day would improve EA in female athletes.

Pre-practice CHO fueling and post-practice CHO and protein intake for recovery and muscle adaptation purposes should be optimized in most athletes. Athletes met CHO intake recommendations before endurance sessions in 66% of the sessions. After intensive training sessions or strength workouts, athletes met protein intake guidelines only in 49% cases. Getting close to 100% compliance for pre- and post-exercise nutrition would likely improve recovery and optimize performance in most participating athletes [29]. The positive correlations between EEE and EI, as well as EA, protein, fat and CHO intake, indicates that during days with a higher EEE a higher EI followed. In a proposed periodized nutrition approach, fueling CHO before a very intensive session might positively influence the quality and overall intensity of the session [17]. Carbohydrate as a fuel after the session would depend on the next session and whether periodized CHO intake was chosen. As we were not able to judge a periodized nutritional approach, we did not analyze post-session CHO intake. Still, CHO intake after a session regarding recovery is an important factor as well. Nevertheless, ingestion of the recommended protein intake after the session should be chosen in any case to optimize recovery and adaptation. This study has only looked for pre-exercise CHO intake and post-exercise protein intake as the two recommendations which should be met in any case. In a future study, a further aim could be to detect whether an athlete follows either a periodized approach or a high-CHO approach during the course of a week. Nevertheless, we think that a periodized CHO-intake might be an ideal solution for athletes with lower energy requirements (e.g., athletes with a spinal cord injury). With a periodized nutrition approach, the athletes would be able to maximize training intensity and recovery without increasing body weight. Furthermore, athletes with SCI would not need to restrict their diet and risk LEA.

### 4.3. Strengths and Limitations

Participants kept the diary for seven consecutive days, which is rather long but necessary to get an insight of a whole week— mimicking a microcycle of training. Furthermore, a 7-day food record showed a lower variability in food intake compared to a food record with a shorter duration (24 h or 3 days) [44]. Capling et al. [14] reported that self-reported food intake needs further development in food quantification (i.e., with photographs). Therefore, athletes in our study weighed their food intake and collected photographs of the meal, which helped to improve the analysis of nutritional intake. The diary might also be limiting, because of its susceptibility to under- or over-reporting of food and training [1,14]. Especially, if we look at Figure 1, we cannot completely exclude under-reporting in F04. Calculating EI:REE according to the Goldberg equation [45] in all athletes resulted in 1.29 ± 0.48 whereas 11 athletes presented a value higher than 1.2. Possibly, under-reporting might have been the case in three athletes. Participants were not familiar with the act of writing down food intake; therefore, healthier meals might have been chosen, due to a higher awareness [14]. Furthermore, estimation of EE during leisure time as well as during training is prone to errors as it is only estimated [1,14]. Additionally, involving athletes with a spinal cord injury seems to be even more of a risk of over- or underestimation of EE as the amount of active muscle mass as well as the lesion itself seem to be influencing factors on EE. Moreover, most studies used for EEE calculation were performed with male athletes, which could also lead to an error of estimation [46].

In one participant, the REE measurement could not have been performed due to technical issues and therefore, REE had to be estimated according to an estimation equation used in a study in tetraplegic athletes [19]. This estimation has not been used for the calculation of EA and, therefore, EA should not have been affected by this limitation.

To study the health consequences of LEA in wheelchair athletes was not our aim. Nevertheless, the investigation of hormones, blood parameters, bone mineral density and other health parameters is important to study in the future. Moreover, the heterogeneity of the athletes (i.e., completeness and level of the injury) prevents an accurate comparison between these different wheelchair athletes. Future multicenter studies should investigate this issue in a higher number of athletes to be able to make comparisons between different lesion levels, genders as well as different sports. Additionally, they should track body composition pre- and post-investigation to assist with the interpretation of under- or over-reporting. Furthermore, the small sample size in this study might not be representative for a true population prevalence. Future studies are needed to draw any conclusions on the prevalence of LEA. 

## 5. Conclusions

A very high amount (13 out of 14) of the athletes in this study showed a suboptimal EA and half of them showed LEA pre-season. Therefore, we suspect that wheelchair athletes and probably other para-athletes as well might be at a risk for LEA. Additionally, a difference among gender was found in this study, with female athletes being more susceptible to LEA. Therefore, further studies on EA in different para-athletic cohorts need to be conducted to investigate the effects of LEA on different health parameters. More importantly, future research needs to focus on cut-off values for optimal, suboptimal, as well as low EA and whether these are the same for para-athletes compared to able-bodied athletes. In addition, tracking body composition and EA during several different time points during a season would be of interest in order to be able to exclude LEA as an issue in this study because of the seasonal phase (i.e., pre-season).

From a fueling perspective, the athletes participating in this study showed some potential in terms of improving their nutritional intake before and after training sessions in order to optimize training and recovery. In general, guidelines for macronutrient intake in wheelchair athletes might be of interest in order to improve individual counseling. Therefore, further research is needed to investigate guidelines for pre-exercise carbohydrate intake and post-exercise protein and carbohydrate intake in wheelchair athletes with a lower amount of active muscle mass. 

## Figures and Tables

**Figure 1 nutrients-12-03262-f001:**
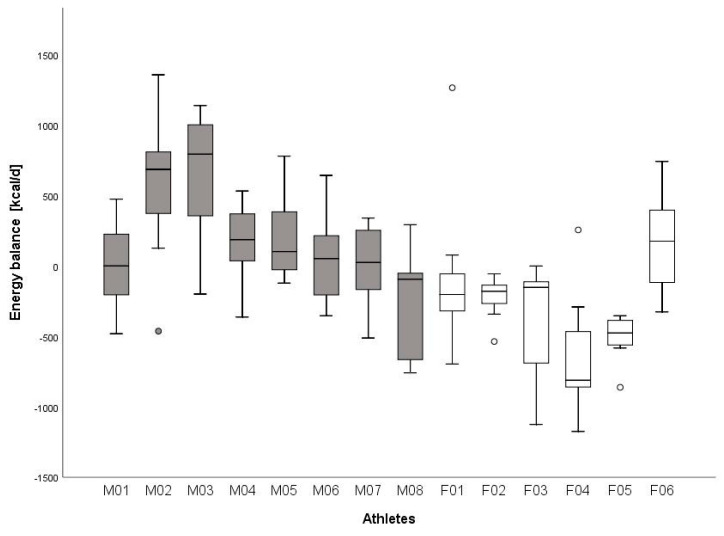
Summary of the daily energy balance for each participant Note: Mxx = male athletes, Fxx = female athletes; o = outlier; data from 7 days included per participant; data presented as the median and the interquartile range.

**Figure 2 nutrients-12-03262-f002:**
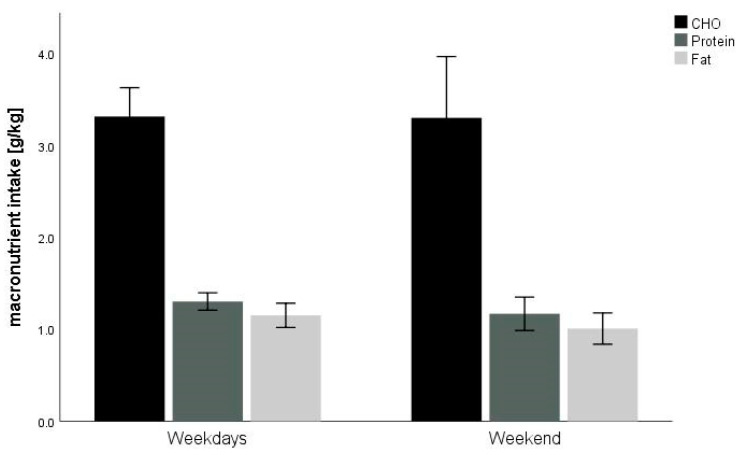
Macronutrient intake on weekdays versus weekends. Note: CHO = carbohydrate; no significant difference between weekdays’ and weekends’ macronutrient intake.

**Table 1 nutrients-12-03262-t001:** Exercise energy expenditure for different sports and intensities.

	Energy Expenditure (kcal kg^−1^ min)			
Sport	Recovery	GA 1	GA 2	Intensive/Race
Handcycling or arm cranking	2.8	4.4	6.0	9.3
Wheelchair racing	2.4	4.3	6.1	7.9
Badminton/tennis session	1.6	2.5	3.3	4.1
	Shooting			Game
Basketball session	3.2			6.1
	Resistance			Circuit
Strength training session	2.2			2.3

Note. GA = Basic endurance, adapted from Conger and Bassett (2011).

**Table 2 nutrients-12-03262-t002:** Anthropometric and disability characteristics.

	Participants		
	All (*n* = 14)	Male (*n* = 8)	Female (*n* = 6)
Age [years]	34.9 ± 9.4	35.8 ± 7.6	33.8 ± 12.1
Height [m]	1.71 ± 0.14	1.80 ± 0.08	1.59 ± 0.10 *
Body mass [kg]	64.9 ± 12	66.5 ± 12.1	62.8 ± 12.6
Body-mass index [kg m^−2^]	22.3 ± 4.4	20.4 ± 2.7	24.9 ± 5.1
Fat-free mass [kg]	46.2 ± 9.7	51.7 ± 8.8	38.9 ± 4.9 *
Fat mass [kg]	18.7 ± 2.3	14.8 ± 4.2	23.9 ± 7.9 *
REE [kcal d^−1^]	1368 ± 259	1506 ± 222	1184 ± 185 *
Disability			
Years since injury	19.6 ± 7.8	18.4 ± 6.8	21.2 ± 9.5
Paraplegia	11 (79%)	6 (75%)	5 (83%)
Tetraplegia	2 (14%)	2 (25%)	0 (0%)
AIS A	9 (64%)	5 (63%)	4 (67%)
AIS B-D	4 (29%)	3 (38%)	1 (17%)

Note. AIS = American Spinal Injury Association (ASIA) Impairment Score, REE = Resting energy expenditure, data are presented as means ± SD or *n* (%). * = significantly different compared to male athletes (*p* < 0.05).

**Table 3 nutrients-12-03262-t003:** Energy availability for each day per athlete over the seven consecutive days.

Participant	Sport	FFM (kg)	EA Day No. 1	EA Day No. 2	EA Day No. 3	EA Day No. 4	EA Day No. 5	EA Day No. 6	EA Day No. 7	Weekly Mean of EA	No. of Days of LEA
M01	Badminton	52.5	*41.3*	*31.7*	*41.8*	*33.4*	**22.3**	**27.0**	**29.3**	*32.4 ± 7.2*	3
M02	Paracycling	57.6	*37.9*	52.8	**28.8**	*41.3*	**17.2**	*41.2*	*44.4*	*37.7 ± 11.5*	2
M03	Paracycling	46.6	55.1	54.1	61.0	58.8	*40.1*	50.0	*33.0*	50.3 ± 10.2	0
M04	Paracycling	36.5	*32.4*	51.0	*38.1*	**23.6**	50.1	*39.4*	*40.0*	*39.2 ± 9.6*	1
M05	Paracycling	47.4	*32.8*	49.0	**26.9**	*37.9*	*32.4*	*42.8*	**29.3**	*35.9 ± 7.8*	2
M06	Athletics	53.9	*41.7*	**23.6**	**29.5**	*33.5*	**22.9**	**28.0**	*36.2*	*30.8 ± 6.8*	4
M07	Paracycling	52.4	**25.5**	*34.2*	*38.9*	**28.2**	*40.0*	*33.0*	*37.3*	*33.9 ± 5.4*	2
M08	Basketball	66.6	*36.6*	*37.8*	**20.4**	*36.1*	**17.3**	*33.4*	**19.7**	**28.8 ± 9.1**	3
Male athletes										36.1 ± 6.7 *	30.4%
F01	Badminton	44.7	**28.1**	*33.3*	**12.7**	**26.3**	64.3	**23.2**	**21.9**	**30.0 ± 16.4**	5
F02	Badminton	42.0	**24.8**	**19.7**	**14.8**	**20.8**	**20.2**	**10.9**	**20.5**	**18.8 ± 4.5**	7
F03	Athletics	30.3	*38.9*	**19.8**	*43.4*	*42.3*	**3.9**	**19.1**	*n*/a	**27.9 ± 16.0**	3
F04	Athletics	37.9	*43.6*	**16.1**	**10.2**	**24.4**	**17.6**	**6.6**	**−0.7**	**16.8 ± 14.3**	6
F05	Tennis	39.5	**19.7**	**26.1**	**12.7**	**20.1**	**23.8**	**24.4**	**26.5**	**21.9 ± 4.9**	7
F06	Paracycling	39.2	49.4	*36.6*	*39.7*	*32.7*	*44.2*	**22.8**	**21.9**	*35.3 ± 10.3*	2
Female athletes										25.1 ± 7.1	73.2%
All athletes										31.4 ± 8.7	48.6%

Note. FFM = Fat-Free Mass [kg], LEA = Low Energy Availability [kcal kg^−1^ FFM], EA = Energy Availability [kcal kg^−1^ FFM], white shading = optimal EA (≥45 kcal kg^−1^ FFM day^−1^), italics numbers= reduced EA (between 30 and 45 kcal kg^−1^ FFM day^−1^), bold numbers = LEA (≤30 kcal kg^−1^ FFM day^−1^), data are presented as total numbers, means ± SD or percentages; * = significantly different compared to male athletes (*p* < 0.05).

**Table 4 nutrients-12-03262-t004:** Energy expenditure, energy intake and energy availability.

	Participants		
Energy Balance	All (*n* = 14)	Male (*n* = 8)	Female (*n* = 6)
Weekly training [h]	13.0 ± 4.9	13.3 ± 6.4	12.8 ± 2.2
EEE [kcal·d^−1^]	423 ± 142	440 ± 152	400.0 ± 138
EI [kcal·d^−1^]	1891 ± 574	2276 ± 368	1377 ± 337 *
TEE [kcal·d^−1^]	1918 ± 366	2107 ± 355	1666 ± 196 *
Energy balance [kcal·d^−1^]	−27 ± 376	169 ± 305	−289 ± 305 *
EA classified (*n* (%))			
>45 kcal kg^−1^ FFM day^−1^	1/14 (7%)	1/8 (13%)	0/6 (0%)
30–45 kcal kg^−1^ FFM day^−1^	7/14 (50%)	6/8 (75%)	1/6 (17%)
<30 kcal kg^−1^ FFM day^−1^	6/14 (43%)	1/8 (13%)	5/6 (83%)

Note. EEE = Exercise Energy Expenditure, EI = Energy Intake, TEE = Total Energy Expenditure, EA = energy availability, FFM = Fat-Free mass, data are presented as means ± SD or *n* (%), * = significantly different compared to male athletes (*p* < 0.05).

**Table 5 nutrients-12-03262-t005:** Dietary intake before and after training sessions.

	Participants		
Total Daily Macronutrient Intake	All (*n* = 14)	Male (*n* = 8)	Female (*n* = 6)
Carbohydrate, g per day	211 ± 70	260 ± 16	146 ± 13 *
g kg^−1^ BM	3.3 ± 1.2	4.0 ± 0.3	2.4 ± 0.3 *
Protein, g per day	82 ± 17	91 ± 5	70 ± 6 *
g kg^−1^ BM	1.3 ± 0.2	1.4 ± 0.4	1.1 ± 0.1 *
Fat, g per day	72 ± 27	86 ± 8	52 ± 8 *
g kg^−1^ BM	1.1 ± 0.4	1.3 ± 0.1	0.9 ± 0.2 *
**Number of days, where guidelines for total daily protein and carbohydrate intake were met (*n* (%))**	**All (*n* = 97 days)**	**Male (*n* = 56 days)**	**Female (*n* = 41 days)**
**Guidelines**	**No. days**	**No. days**	**No. days**
Protein intake ≥ 1.2 g kg^−1^ BM	55 (57%)	39 (70%)	16 (39%)
Carbohydrate intake ≥ 3 g kg^−1^ BM	55 (57%)	46 (82%)	9 (22%)
**Number of sessions, where pre- and post-training guidelines were met (*n* (%))**	**All (*n* = 74 sessions)**	**Male (*n* = 40 sessions)**	**Female (*n* = 34 sessions)**
**Guidelines**	**No. sessions**	**No. sessions**	**No. sessions**
Protein intake: 20–25 g after intensive or strength training session	36 (49%)	22 (55%)	14 (41%)
Carbohydrate intake: 1–4 g kg^−1^ BM before endurance training session	49 (66%)	25 (63%)	24 (71%)

Note. BM = Body mass, data are presented as means ± SD or *n* (%), Sport nutritional guidelines according to Broad (2019). * = significantly different compared to male athletes.

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
