# Peer review of "Energy Availability in Male and Female Elite Wheelchair Athletes over Seven Consecutive Training Days"

_nutrients, 2020, doi:10.3390/nu12113262_

Round 1

Reviewer 1 Report

Main Comments:

This paper is novel in that it examined energy availability of elite wheelchair athletes. This information is warranted given that we have little information or studies on this topic. I commend the authors for conducting what is not an easy study design, especially when it comes to measuring energy availability. However, the organization and writing of this manuscript needs significant attention.

Major comments

Abstract:

Lines 20-21: the authors may need to change the % of athletes at risk of LEA if using a different threshold.

Line 22: Suggest a stronger conclusion to summarize findings… “LEA was predominantly what”???

Introduction:

First paragraph- I would restate this to note that RED-S is an expansion of the Triad and is not intended to replace the Triad. Explain that it also includes males

Second Para- please better define the LEA calculation (see Loucks work). Is Mountjoy the original reference for this too?  Line 42 is similar and repetitive. Suggest combining these paragraphs or reorganize for clarity

Line 44: EA at or below 30? , what about males? Same LEA threshold?

Line 53-57: Reword this. The wording is confusing…What about the RED-S assessment tool? Can we look at BMD and hormones?

Line 88: Did ALL athletes get support from the Sports Nutritionist? Seems like this just be standardized

For clarification purposes, is it appropriate to say the athletes were “at risk” for LEA or had LEA? See other literature. I realize that this was based on “calculated EA”, but does that determine risk or actual LEA?

DXA procedures – provide more detail. Why was BMD not included with its relevance to EA?

Lines 130-140: The authors need to provide detail on how EA was calculated. Did they subtract for RMR during the EEE calculation? See Heikura 2018 for details on this

Why include TEF? Do other papers examining LEA calculate this?

Line 140: “the EA was described elsewhere”—do the authors use the same procedures for measuring EA in the review paper as this paper? Provide more detail on how EA was calculated as this is the primary purpose of the paper.

Line 145: maybe a different subtitle to describe this section and a bit more detail on the pre and post exercise period. How was this defined? Anything consumed immediately after or up to 2hours post? The results show pre intake but this is not mentioned in the methods either

Line 164-166: This doesn’t make sense. Are the authors referring to EI or EA here?

Table 3: is it better to represent data as #/7 days in LEA vs %? As a % may be misleading?

Not sure you can mean the males and females if using a diff LEA threshold.

Why did the authors choose the same LEA threshold for males? See other literature (Heikura et al. 2018, and Fagerberg 2017) suggesting a lower threshold of 20-25 kcal/kg FFM in male athletes.

Results- the authors may need to change the % of male athletes with LEA if using the male threshold

Is it appropriate to use a % to describe the population of athletes since it was a small N? Suggest expressing as a number (ie 7 out of 10…etc)

Figure 1 & 2 – nice figure but delete gridlines

Figure 2- what does the ”y-axis” legend say? Rel. intake to what? how about “macronutrient intake”

Table 3: the top row of descriptors is messy and needs to be reformatted.

Discussion- the authors may need to rework some of the findings with the male threshold for LEA in mind.

Line 191-193- reword this to say something similar to “ female athletes displayed LEA on 73% of the days…” and use the same on other sentences.

Lines 200- 205: The use of % as a descriptor in sentences needs to be rewritten for clarification purposes throughout the paper…

Line 211: Can you use a % given that this is a small N to describe your population?

Line 248/249: CHO fueling is also important for recovery? Arguably more so than protein. Why do the authors not include this? Suggest including an analysis of Carbohydrate intake too (this is  mentioned in the methods too?)

Lines 250-252. Reword these as this use of % is confusing in the text.

Line 256-257- please clarify… again why the emphasis on only protein after the training session as the ACSM/AND recs are different to what the authors are suggesting

Line 263: pLease explain. Are the authors suggesting that CHO intake post exercise may help restore EA??

Limitations

Would the small sample size not be a limitation and suggest that this may not be representative of a “true” prevalence of LEA in this population? And perhaps that this paper is seeking to establish methods of determining LEA/risk in this population?

Conclusion- again not sure that the authors can truly use a % to describe the prevalence—is this truly representative with a small sample size? Might be better to use N/14

Reword the second sentence to say that wheelchair athletes are “at risk”…

Line 304- reword to say “From a fueling perspective…”

The authors need to rework the last few sentence on the conclusion-Why this emphasis on only protein after exercise? It’s important to replace muscle glycogen post exercise.

Minor comments

Abstract

Line 12: Use “Energy Availability (EA)” vs the availability of energy… Also add the acronym because its use later in the abstract but not defined & reword the rest of the sentence so the reader knows its energy intake- exercise energy expenditure…..

Line 14: How often? Do we have a %?

Line 14: use “aesthetic sports” vs. weight sensitive

Line 15: Add “elite” descriptor to the subject group

Line 17&18: combine in to 1 sentence

Line 19: from all “athlete days”? Do you mean “training days”

Line 23: delete “their”

Line 28: change intensively to “extensively”

Intro:

Line 66:  reword to: “Therefore, research is warranted to examine EA in athletes with SCI…..”

Line 67-69: Perhaps move to discussion after finding of this study?

Methods:

Line 78: reword “between the ages of 18-60…”

Line 79: reword: “athletes were excluded”

Line 102: Participants were measured? OR Oxygen/ CO2 was measured? Be specific here

Line 112: To obtain “normal” glycogen stores?

Line 116: Same question as above… did all participants receive training from the sport nutritionist on completing diet records?

Line 125: BORG what? Give more description

Results/Discussion:

Is “g” the correct denotation for effect size? Is it not “ES”?

Table 2: why the line above and below the REE category? Perhaps just below REE would better separate the data

Line 163: were in LEA? Or displayed/at risk for LEA? – clean up the writing on this sentences

Line 163: is it best to describe as #/7 days vs using a % ?

Line 216: change loose to “lose”

217: change to “in pre-season”

Line 224: reference the Fig # not boxplot

Line 238: change nutritional to “macronutrient”

Line 241: insert the “reported” intakes

Line 246: change “to ensure optimal health and performance”

Line 266: change “they” to “athletes with SCI or…”

Line 269: delete “usually”

Author Response

Major comments

Abstract:

Lines 20-21: the authors may need to change the % of athletes at risk of LEA if using a different threshold.

Thank you for this comment. We did not adjust the threshold for men as further outlined below.

Line 22: Suggest a stronger conclusion to summarize findings… “LEA was predominantly what”???

Thank you for this comment. This was changed accordingly.

Introduction:

First paragraph- I would restate this to note that RED-S is an expansion of the Triad and is not intended to replace the Triad. Explain that it also includes males

Thank you for this input. We changed this paragraph following your suggestion.

Second Para- please better define the LEA calculation (see Loucks work). Is Mountjoy the original reference for this too?  Line 42 is similar and repetitive. Suggest combining these paragraphs or reorganize for clarity

Thank you for this comment. This was adapted according your suggestion.

Line 44: EA at or below 30? , what about males? Same LEA threshold?

This was changed accordingly. As there is no other threshold for LEA in male athletes, this was not changed.

Heikura et al. 2018 cited this article (Fagerberg 2017) but still they used the same threshold for the interpretation for EA in male and female athletes. As there is no general consent on the implementation of this lower threshold, we leave our analysis and interpretation based on the "general" values as done by Heikura et al. 2018. Possibly in the future, when a general consent was made about different thresholds for different population groups (e.g. male, female, SCI), one could implement a different interpretation strategy. Until then, using different cut-off values would be confusing also in the comparison with other research done on this topic.

Line 53-57: Reword this. The wording is confusing…What about the RED-S assessment tool? Can we look at BMD and hormones?

This was changed accordingly.

Line 88: Did ALL athletes get support from the Sports Nutritionist? Seems like this just be standardized

The wording has been changed to make it more clear for the reader.

For clarification purposes, is it appropriate to say the athletes were “at risk” for LEA or had LEA? See other literature. I realize that this was based on “calculated EA”, but does that determine risk or actual LEA?

Thank you for this comment. It seems extremely difficult to give an exact answer to this question. On one hand, there are clear threshold values available to interpret energy availability. Outlining the limitations of the calculations, we just interpreted energy availability just by these thresholds. On the other side, it seems already known, that those exact thresholds might not be the "best practice" in the clinical application. A more dense assessment including different outcome parameters (bone mineral density, REE, hormones) seems to be more appropriate to fully "diagnose" LEA. Nevertheless, for reason of simplicity, we stay with the term "showed LEA" by including the interpretation of the thresholds.

DXA procedures – provide more detail. Why was BMD not included with its relevance to EA?

Thank you for this comment. A full-body DXA measurement has been performed to measure body composition. A second measurement would have been undertaken, if we would have needed bone mineral density at a special site (e.g. spine or femoral os). This was not our primary goal. In addition, we already now, that bone mineral density is further diminished by the inactivity of the legs and by the daily wheelchair use in this population. Most athletes show therefore a very low bone mineral density resulting in osteopenia or even osteoporosis. Therefore, the additional analysis of bone mineral density would not have resulted in the expected information in addition to the calculation of EA.

Lines 130-140: The authors need to provide detail on how EA was calculated. Did they subtract for RMR during the EEE calculation? See Heikura 2018 for details on this

Thank you for this suggestion. We calculated the EEE based on the values from Conger and Basset 2011. They describe the compendium of physical activity for wheelchair users. In their manuscript they describe, that due to an altered oxygen consumption in wheelchair exercise, they do not describe the energy costs of activities using METs. They describe it by using kcal per kg per min. We felt it would not be appropriate to do it the same way in individuals with SCI.

Here is what Conger and Basset (2011) write: "Previous compendiums have reported the energy costs of activities expressed as METs. This can be a source of confusion in that the literature has reported one MET as the fixed value of 3.5 ml of oxygen · kg body weight-1 · min -1 (Balke, 1960) and as the measured resting metabolic rate (RMR; Dill, 1936). Acknowledging that the RMR of some populations of individuals who use wheelchairs is not equal to 3.5 ml·kg-1·min-1 and that there is some controversy as to which definition of a MET is correct, we chose to express the energy cost of activities as kcal·kg-1·hr-1."

Why include TEF? Do other papers examining LEA calculate this?

We are grateful for this question. As we did also calculate TEE and energy balance, we calculated TEF as well. This is now clearly stated in the manuscript.

Line 140: “the EA was described elsewhere”—do the authors use the same procedures for measuring EA in the review paper as this paper? Provide more detail on how EA was calculated as this is the primary purpose of the paper.

Thank you for this suggestion. The calculation is now added to make it more clear.

Line 145: maybe a different subtitle to describe this section and a bit more detail on the pre and post exercise period. How was this defined? Anything consumed immediately after or up to 2hours post? The results show pre intake but this is not mentioned in the methods either

Thank you for your great comment. We have added in the food diary a specific sentence to make it more clear, that around training nutrition was analysed as well. Furthermore, the two terms "pre" and "post" training have been specified to make it more clear for the reader.

Line 164-166: This doesn’t make sense. Are the authors referring to EI or EA here?

This sentence has been deleted.

Table 3: is it better to represent data as #/7 days in LEA vs %? As a % may be misleading?

Thank you for this suggestion. This has been changed accordingly.

Not sure you can mean the males and females if using a diff LEA threshold.

As explained below, we did not use different LEA threshold.

Why did the authors choose the same LEA threshold for males? See other literature (Heikura et al. 2018, and Fagerberg 2017) suggesting a lower threshold of 20-25 kcal/kg FFM in male athletes.

This is correct, that Fagerberg 2017 suggested a lower threshold in male athletes. Heikura et al. 2018 cited this article but still they used the same threshold for the interpretation for EA in male and female athletes. As there is no general consent on the implementation of this lower threshold, we leave our analysis and interpretation based on the "general" values as done by Heikura et al. 2018. Possibly in the future, when a general consent was made about different thresholds for different population groups (e.g. male, female, SCI), one could implement a different interpretation strategy. Until then, using different cut-off values would be confusing also in the comparison with other research done on this topic.

Results- the authors may need to change the % of male athletes with LEA if using the male threshold

See comment above.

Is it appropriate to use a % to describe the population of athletes since it was a small N? Suggest expressing as a number (ie 7 out of 10…etc)

This has been changed.

Figure 1 & 2 – nice figure but delete gridlines

They have been deleted.

Figure 2- what does the ”y-axis” legend say? Rel. intake to what? how about “macronutrient intake”

This has been changed according to your suggestion.

Table 3: the top row of descriptors is messy and needs to be reformatted.

This has been reformatted.

Discussion- the authors may need to rework some of the findings with the male threshold for LEA in mind.

N/A

Line 191-193- reword this to say something similar to “ female athletes displayed LEA on 73% of the days…” and use the same on other sentences.

This was changed.

Lines 200- 205: The use of % as a descriptor in sentences needs to be rewritten for clarification purposes throughout the paper…

This has been changed accordingly.

Line 211: Can you use a % given that this is a small N to describe your population?

In this case, we feel it is appropriate as it includes all 7 days for all 14 participants. In the other cases were % for the 6 female or 8 male participants were given, we adapted to one out of 6 or one out of 8.

Line 248/249: CHO fueling is also important for recovery? Arguably more so than protein. Why do the authors not include this? Suggest including an analysis of Carbohydrate intake too (this is  mentioned in the methods too?)

It seems not unusual to undertake a periodized nutritional approach or a specific training session under low carbohydrate availability, we felt it would be too complicated to analyse post-session CHO refueling. Of course, this can also compromise recovery for the subsequent session but dividing between "low carbohydrate availability" and "bad nutrition habits" in terms of CHO refuelling would be complicated. Therefore, we decided to not include any analysis on that.

Lines 250-252. Reword these as this use of % is confusing in the text.

This has been changed accordingly.

Line 256-257- please clarify… again why the emphasis on only protein after the training session as the ACSM/AND recs are different to what the authors are suggesting

We agree with you. We clarified the reason, why we did no analyse post-session CHO intake and that still this is a factor of concern regarding recovery.

Line 263: pLease explain. Are the authors suggesting that CHO intake post exercise may help restore EA??

As we did not analyse CHO intake post exercise, we do not suggest this. But of course, looking at the total amount of CHO ingested by the female athletes, it might be expected, that a higher CHO intake over the whole day would help to improve EA in these athletes.

Limitations

Would the small sample size not be a limitation and suggest that this may not be representative of a “true” prevalence of LEA in this population? And perhaps that this paper is seeking to establish methods of determining LEA/risk in this population?

Thank you for this comment. This has been added to the limitations section.

Conclusion- again not sure that the authors can truly use a % to describe the prevalence—is this truly representative with a small sample size? Might be better to use N/14

This has been changed.

Reword the second sentence to say that wheelchair athletes are “at risk”…

This has been changed accordingly.

Line 304- reword to say “From a fueling perspective…”

Changed accordingly.

The authors need to rework the last few sentence on the conclusion-Why this emphasis on only protein after exercise? It’s important to replace muscle glycogen post exercise.

we agree, this has been changed accordingly.

Minor comments

Abstract

Line 12: Use “Energy Availability (EA)” vs the availability of energy… Also add the acronym because its use later in the abstract but not defined & reword the rest of the sentence so the reader knows its energy intake- exercise energy expenditure…..

This was changed accordingly.

Line 14: How often? Do we have a %?

As this includes several different sports and gender, we do not feel that is appropriate to give any specific %-value for an overall estimation of the prevalence of LEA.

Line 14: use “aesthetic sports” vs. weight sensitive

This was added.

Line 15: Add “elite” descriptor to the subject group

This has been changed.

Line 17&18: combine in to 1 sentence

Changed accordingly.

Line 19: from all “athlete days”? Do you mean “training days”

This has been changed.

Line 23: delete “their”

changed.

Line 28: change intensively to “extensively”

changed.

Intro:

Line 66:  reword to: “Therefore, research is warranted to examine EA in athletes with SCI…..”

Changed following your suggestion.

Line 67-69: Perhaps move to discussion after finding of this study?

This was not moved, as we think it is suitable here.

Methods:

Line 78: reword “between the ages of 18-60…”

changed.

Line 79: reword: “athletes were excluded”

changed.

Line 102: Participants were measured? OR Oxygen/ CO2 was measured? Be specific here

changed.

Line 112: To obtain “normal” glycogen stores?

changed.

Line 116: Same question as above… did all participants receive training from the sport nutritionist on completing diet records?

Yes, this has been added.

Line 125: BORG what? Give more description

This has been added.

Results/Discussion:

Is “g” the correct denotation for effect size? Is it not “ES”?

We calculated the Hedges g and not ES (according to another reviewer comment).

Table 2: why the line above and below the REE category? Perhaps just below REE would better separate the data

Changed.

Line 163: were in LEA? Or displayed/at risk for LEA? – clean up the writing on this sentences

this has been changed.

Line 163: is it best to describe as #/7 days vs using a % ?

N/A

Line 216: change loose to “lose”

changed.

217: change to “in pre-season”

changed.

Line 224: reference the Fig # not boxplot

changed.

Line 238: change nutritional to “macronutrient”

changed.

Line 241: insert the “reported” intakes

changed accordingly.

Line 246: change “to ensure optimal health and performance”

changed.

Line 266: change “they” to “athletes with SCI or…”

changed.

Line 269: delete “usually”

deleted.

Reviewer 2 Report

Thanks you for the opportunity to review this study. Research in para-wheel-chair athletes is need to help guide practice of performance nutritionists. This manuscript is of relevance and would be a positive addition to the evidence base of sports nutrition in para-athletes. However, there are concerns around some of the methodology including the statistics employed as well as numerous grammatical and spelling errors that would need to be addressed in order to strengthen the manuscript.

Abstract:

The background section of the abstract is long and detailed. The definition of energy availability is not necessarily needed in the abstract. The extra words could be used to provide more detail in the methods section for example “seven day food diary to estimate energy expenditure and exercise energy expenditure”.

L21 – please add a space between the numbers and the ± sign

L22 – the first sentence in this line is incomplete.

Introduction

This section provides good background and justification for the study, however there a several grammatical error, including tense, affecting flow of writing, while some paragraphs are repeated. Please see below some specific comments and some examples of where grammar/spelling should be corrected.

L39 – sentence flow, suggest deleting “…FFM regarding…”  

L42 – This sentence is too similar to the definition in L37, and is repetitive. Suggest re-writing L37 – 46 so that definition is only stated once.

L42  - 46 – switch “were“ and “was” to “are” and “is” to keep tense consistent as present tense.

L47 – Introduce SCI abbreviation here rather than in L49.

L 49 – delete “spinal cord injury” and keep abbreviation.

L52 – swap “adaptation” to “adaptive”

L56 – 57. Are you please able to explain what you mean in this sentence? It’s the use of the word “during” that is causing confusion.

L58-59 – Can you give an example of “at the major competition”? Only examples that come to mind for short time-reduced EA would be athletes who have to make weight, which isn’t a change of body composition, but rather a change of body mass through acute manipulation of food intake and fluid.

L66 – tense ‘have’ to ‘has’

L71 – delete comma. Swap ‘to develop’ to ‘of developing’.

Methods:

This section is generally well written except for some minor grammar and spelling mistakes. It provides good detail in how the study was conducted, however, there are questions regarding DXA methodology.  A major concern is the suitability of the statistics used for the design of this study.

L79 – swap ‘have been’ for ‘were’

L80 move comma from after ‘excluded’ to after ‘ dietary intake’.

L99-106 – was the energy expenditure for RMR calculated by the metabolic cart software or was EE determined using an equation using the steady state VO2 and VCO2 Ve values? If an equation was used please provide a reference for it.

Please provide the error of measurement for the metabolic cart.

L107: Please provide DXA analysis software and version. Are you please able to provide the error of the machine for FM and FFM. The technician who conducted the scans, did this include scan analysis or just the position and scanning of the participants?

L110 – what was the time frame for the drink of water beforehand and the scan time? Were all participants provided the same volume of water, was the volume adjusted for body mass? If it was provided just before the scan it is not going to hydrate the athlete and instead will likely show as lean mass in the trunk and thus affect the measurement of FFM? The Best Practice protocol cited actually recommends no fluid intake on the morning of the scan.

L116 – swap “have been” for “were”

L125 – Please state “rate of perceived of Exertion (RPE)” after Borg

L126 – what are the four intensities? Are these stated anywhere in the text?

L136 – Please write out actual Author name before reference.

Statistics

As this study contains multiple time points for variables EI, EE, EA and EB, a t-test to assess the difference between males and females does not seem appropriate. Wouldn’t a a repeated measures ANOVA would be better suited?

Results and Discussion

The results and conclusions drawn regarding EA, EE and EI comparisons may have to be amended or withdrawn if data is re-analysed with appropriate statistics. 

Table 4

Please remove the decimal place from EEE, EI and TEE.

The “mean EA over week” section of Table 4 feels disjointed from the information above and should be in its own separate table.

Figure 1: Please state how data is presented in the title. Is it means or medians?

Figure 2: It states no significant difference, but does not clarify the no difference between what? between Protein and fat intake or between weekdays and weekend?

Table 5

Similar to Table 4 the two sections of this table do not connect well. “Number of days, where guidelines for total”… should be in its own separate table.

Discussion:

L207 – underreporting is suggested as a reason for the high number of LEA in the female athletes a number of times in this manuscript. Was there any check for for under-reporting bias? For instance Goldberg et al. 1991 or Black 2000.

L211 – 212 Please reword this sentence as it is not clear what is being said.

L216 – “lose” not “loose”.

L223 -  “median” is mentioned here for the first time. Figure 1 title does not state that the data shown is Median and interquartile range. In the statistics section it states all data are presented as mean ± SD. If data in Figure 1 are presented differently it needs to be advised in the figure title.

L225 – It is stated that females were in negative energy balance over all seven days. Would it have been possible to record body mass over the study period in support of this?

L243 – It has been mentioned before about athletes restricting food intake due to “intended pre-season weight loss”. However, were the athletes asked if that was what they were doing? It reads like this is assumed to be the case. LEA can be due to a high exercise energy expenditure and not just low energy intake. Could it be that the athletes were unaware or didn’t have sufficient knowledge / education, and that the low energy intake was not intentional?

L249-250 – please add “did” before “…CHO intake”. I also think you’re missing the word “meet” before recommendations.

L258 – word choice, “should” not “would”.

L271 – when citing in text it should be “Capling et al”

L279 – there are some words missing in this sentence. For it to make sense it should read “…seems to be even more of a risk of…”

For future studies would assessing body mass at the end of the 7 days assist with interpreting if athletes had over or under estimated food intake?

L292: Multicentre studies would also enable comparisons between different sports (e.g. sprint track, endurance, rugby, basketball etc)

Conclusions:

L295 – delete “might”

L300 – add “and” before “whether these…“.

References

Goldberg, G.R., Black, A.E., Jebb, S.A., Cole, T.J., Murgatroyd, P.R. Coward, W.A. and Prentice, A.M. (1991) Critical evaluation of energy intake data using fundamental principles of energy physiology: 1. Derivation of cut-off limits to identify under-recording. European Journal of Clinical Nutrition 45, 569-581.

Black, A. E. (2000). Critical evaluation of energy intake using the Goldberg cut-off for energy intake: Basal metabolic rate. A prac- tical guide to its calculation, use and limitations. International Journal of Obesity, 24(9), 1119–1130. doi:10.1038/sj.ijo. 0801376

Author Response

Abstract:

The background section of the abstract is long and detailed. The definition of energy availability is not necessarily needed in the abstract. The extra words could be used to provide more detail in the methods section for example “seven day food diary to estimate energy expenditure and exercise energy expenditure”.

This has been changed.

L21 – please add a space between the numbers and the ± sign

This was changed accordingly.

L22 – the first sentence in this line is incomplete.

The whole sentence was changed.

Introduction:

This section provides good background and justification for the study, however there a several grammatical error, including tense, affecting flow of writing, while some paragraphs are repeated. Please see below some specific comments and some examples of where grammar/spelling should be corrected.

L39 – sentence flow, suggest deleting “…FFM regarding…” 

this sentence has been changed.

L42 – This sentence is too similar to the definition in L37, and is repetitive. Suggest re-writing L37 – 46 so that definition is only stated once.

This has been changed.

L42  - 46 – switch “were“ and “was” to “are” and “is” to keep tense consistent as present tense.

This has been changed.

L47 – Introduce SCI abbreviation here rather than in L49.

changed.

L 49 – delete “spinal cord injury” and keep abbreviation.

changed.

L52 – swap “adaptation” to “adaptive”

changed.

L56 – 57. Are you please able to explain what you mean in this sentence? It’s the use of the word “during” that is causing confusion.

This has been deleted.

L58-59 – Can you give an example of “at the major competition”? Only examples that come to mind for short time-reduced EA would be athletes who have to make weight, which isn’t a change of body composition, but rather a change of body mass through acute manipulation of food intake and fluid.

Yes of course you are right. The term "e.g. the last few weeks" has been added to clarify.

L66 – tense ‘have’ to ‘has’

We left "have" as we meant two points (occurrence and health consequences).

L71 – delete comma. Swap ‘to develop’ to ‘of developing’.

Changed.

Methods:

This section is generally well written except for some minor grammar and spelling mistakes. It provides good detail in how the study was conducted, however, there are questions regarding DXA methodology.  A major concern is the suitability of the statistics used for the design of this study.

L79 – swap ‘have been’ for ‘were’

changed.

L80 move comma from after ‘excluded’ to after ‘ dietary intake’.

changed.

L99-106 – was the energy expenditure for RMR calculated by the metabolic cart software or was EE determined using an equation using the steady state VO2 and VCO2 Ve values? If an equation was used please provide a reference for it.

this was added in the methods section.

Please provide the error of measurement for the metabolic cart.

Inserted.

L107: Please provide DXA analysis software and version. Are you please able to provide the error of the machine for FM and FFM. The technician who conducted the scans, did this include scan analysis or just the position and scanning of the participants?

This has been added. Furthermore a study by Keil et al. 2016 showed the measurement precision in elite wheelchair athletes.

L110 – what was the time frame for the drink of water beforehand and the scan time? Were all participants provided the same volume of water, was the volume adjusted for body mass? If it was provided just before the scan it is not going to hydrate the athlete and instead will likely show as lean mass in the trunk and thus affect the measurement of FFM? The Best Practice protocol cited actually recommends no fluid intake on the morning of the scan.

A lot of wheelchair athletes need to take daily medication. Therefore, not drinking a glass of water in the morning is very difficult for them. Thus, they were allowed to have 1 glass of water. As this was done in all measurements, it was kind of standardized. But we agree, that if we want to standardize even more precisely, hydration status should be checked before the measurement. But again, in athletes using urine collection bags, this seems to be very difficult.

L116 – swap “have been” for “were”

changed.

L125 – Please state “rate of perceived of Exertion (RPE)” after Borg

changed.

L126 – what are the four intensities? Are these stated anywhere in the text?

This was added in the text and displayed in Table 1 as well.

L136 – Please write out actual Author name before reference.

added.

Statistics:

As this study contains multiple time points for variables EI, EE, EA and EB, a t-test to assess the difference between males and females does not seem appropriate. Wouldn’t a a repeated measures ANOVA would be better suited?

Thank you for this comment. As we compared the outcome of one parameter between two groups (e.g. male and female athletes), a t-test seems to be appropriate. We do not have more than two groups to compare.

Results and Discussion:

The results and conclusions drawn regarding EA, EE and EI comparisons may have to be amended or withdrawn if data is re-analysed with appropriate statistics.

Table 4:

Please remove the decimal place from EEE, EI and TEE.

changed.

The “mean EA over week” section of Table 4 feels disjointed from the information above and should be in its own separate table.

Thank you for this comment. This has been changed accordingly. This parameter means how many participants showed a mean EA over the 7 days in a state of LEA, suboptimal EA or optimal EA.

Figure 1:

Please state how data is presented in the title. Is it means or medians?

The boxplot does always present as the median. To clarify this issue, we added the description to the Figure. We have chosen to show the data with a boxplot as this does better reflect the distribution of energy balance over the whole week for each participant.

Figure 2:

It states no significant difference, but does not clarify the no difference between what? between Protein and fat intake or between weekdays and weekend?

This is also stated in the text. Nevertheless, we added a clarification in the Figure description.

Table 5:

Similar to Table 4 the two sections of this table do not connect well. “Number of days, where guidelines for total”… should be in its own separate table.

Thank you for this comment. This has been changed accordingly to make it more clearly to the reader.

Discussion:

L207 – underreporting is suggested as a reason for the high number of LEA in the female athletes a number of times in this manuscript. Was there any check for for under-reporting bias? For instance Goldberg et al. 1991 or Black 2000.

Thank you for this suggestion. We added the calculation in the limitations section. The definition of the cut-off values is difficult. We know from the literature, that individuals with a spinal cord injury show lower PAL values for the same activity (e.g. you push the wheelchair once and it keeps rolling vs. walking). Therefore, judging any PAL value or define PAL values for the population seems very difficult and would open up another discussion. I hope that the inclusion of the calculation the description of the values help to further interpret the presented data.

L211 – 212 Please reword this sentence as it is not clear what is being said.

This was changed.

L216 – “lose” not “loose”.

this was changed.

L223 -  “median” is mentioned here for the first time. Figure 1 title does not state that the data shown is Median and interquartile range. In the statistics section it states all data are presented as mean ± SD. If data in Figure 1 are presented differently it needs to be advised in the figure title.

Thank you for this suggestion. We added that the data is presented as the mean and interquartile range.

L225 – It is stated that females were in negative energy balance over all seven days. Would it have been possible to record body mass over the study period in support of this?

Thank you for this comment. As underlined in the limitations section, a negative energy balance might have occurred due to underreporting. Tracking of body composition throughout the week would have been possible, but was not intended in a first step. We expected athletes not to be in a dietary restriction phase as this was an exclusion criteria. Therefore, we did not intend to track body composition over this seven days. The exclusion criteria has been added to the methods section.

L243 – It has been mentioned before about athletes restricting food intake due to “intended pre-season weight loss”. However, were the athletes asked if that was what they were doing? It reads like this is assumed to be the case. LEA can be due to a high exercise energy expenditure and not just low energy intake. Could it be that the athletes were unaware or didn’t have sufficient knowledge / education, and that the low energy intake was not intentional?

Thank you for this comment. We have added this to the discussion, as you are totally right. Of course it could also be, that they increased training volume without adjusting the energy intake.

L249-250 – please add “did” before “…CHO intake”. I also think you’re missing the word “meet” before recommendations.

Thank you for this comment. The whole sentence was reformulated to make it more clear to the reader.

L258 – word choice, “should” not “would”.

Changed.

L271 – when citing in text it should be “Capling et al”

This was changed.

L279 – there are some words missing in this sentence. For it to make sense it should read “…seems to be even more of a risk of…”

This was changed according your suggestion.

For future studies would assessing body mass at the end of the 7 days assist with interpreting if athletes had over or under estimated food intake?

Thank you for this suggestion. This was added.

L292: Multicentre studies would also enable comparisons between different sports (e.g. sprint track, endurance, rugby, basketball etc)

added.

Conclusions:

L295 – delete “might”

changed.

L300 – add “and” before “whether these…“.

Changed.

References

Goldberg, G.R., Black, A.E., Jebb, S.A., Cole, T.J., Murgatroyd, P.R. Coward, W.A. and Prentice, A.M. (1991) Critical evaluation of energy intake data using fundamental principles of energy physiology: 1. Derivation of cut-off limits to identify under-recording. European Journal of Clinical Nutrition 45, 569-581.

Black, A. E. (2000). Critical evaluation of energy intake using the Goldberg cut-off for energy intake: Basal metabolic rate. A prac- tical guide to its calculation, use and limitations. International Journal of Obesity, 24(9), 1119–1130. doi:10.1038/sj.ijo. 0801376

Round 2

Reviewer 1 Report

The authors have done a nice job updating a paper on an important topic that needs more attention. 

Table 2 - change "years since disability" - is this the correct way to state this? 

Table 3- I see the notes about the "gray shading" but wondering if there is a better way to make this obvious. It took me a while to figure it out

Table 4... maybe more appropriate to use: 1/14 vs "1 out of 14"

Author Response

Table 2 - change "years since disability" - is this the correct way to state this?

We changed it to years since injury as this is the correct way to state it.

Table 3- I see the notes about the "gray shading" but wondering if there is a better way to make this obvious. It took me a while to figure it out

Thank you for this comment. We changed this to make it more obvious for the reader.

Table 4... maybe more appropriate to use: 1/14 vs "1 out of 14"

Thank you for this suggestion, this was changed accordingly.

Reviewer 2 Report

Thank you for the opportunity to review this study once more. The authors have addressed majority of concerns and provided relevant explanations in other cases. Along with fixing the grammar and spelling errors, these changes have strengthened the manuscript greatly, making it easier to read and improving its quality.

My only suggestion would be that to include your explanation for why a glass of water was provided/considered for these athletes prior to the DXA scan. It is a good learning point (myself included) for practitioners who would not normally work with wheelchair athletes.

Author Response

My only suggestion would be that to include your explanation for why a glass of water was provided/considered for these athletes prior to the DXA scan. It is a good learning point (myself included) for practitioners who would not normally work with wheelchair athletes.

Thank you for this suggestion, this has been added.